# Towards a Treatment for Neuroinflammation in Epilepsy: Interleukin-1 Receptor Antagonist, Anakinra, as a Potential Treatment in Intractable Epilepsy

**DOI:** 10.3390/ijms22126282

**Published:** 2021-06-11

**Authors:** Gaku Yamanaka, Yu Ishida, Kanako Kanou, Shinji Suzuki, Yusuke Watanabe, Tomoko Takamatsu, Shinichiro Morichi, Soken Go, Shingo Oana, Takashi Yamazaki, Hisashi Kawashima

**Affiliations:** Department of Pediatrics and Adolescent Medicine, Tokyo Medical University, Tokyo 160-0023, Japan; ishiyu@tokyo-med.ac.jp (Y.I.); kanako.hayashi.0110@gmail.com (K.K.); shin.szk@gmail.com (S.S.); vander-sar_0301@yahoo.co.jp (Y.W.); t-mori@tokyo-med.ac.jp (T.T.); s.morichi@gmail.com (S.M.); soupei59@gmail.com (S.G.); oanas@tokyo-med.ac.jp (S.O.); tyamaz@tokyo-med.ac.jp (T.Y.); hisashi@tokyo-med.ac.jp (H.K.)

**Keywords:** anakinra, cytokine, febrile infection-related epilepsy syndrome, IL-1β, IL-1 receptor antagonist

## Abstract

Febrile Infection-Related Epilepsy Syndrome (FIRES) is a unique catastrophic epilepsy syndrome, and the development of drug-resistant epilepsy (DRE) is inevitable. Recently, anakinra, an interleukin-1 receptor antagonist (IL-1RA), has been increasingly used to treat DRE due to its potent anticonvulsant activity. We here summarized its effects in 38 patients (32 patients with FIRES and six with DRE). Of the 22 patients with FIRES, 16 (73%) had at least short-term seizure control 1 week after starting anakinra, while the remaining six suspected anakinra-refractory cases were male and had poor prognoses. Due to the small sample size, an explanation for anakinra refractoriness was not evident. In all DRE patients, seizures disappeared or improved, and cognitive function improved in five of the six patients following treatment. Patients showed no serious side effects, although drug reactions with eosinophilia and systemic symptoms, cytopenia, and infections were observed. Thus, anakinra has led to a marked improvement in some cases, and functional deficiency of IL-1RA was indicated, supporting a direct mechanism for its therapeutic effect. This review first discusses the effectiveness of anakinra for intractable epileptic syndromes. Anakinra could become a new tool for intractable epilepsy treatment. However, it does not currently have a solid evidence base.

## 1. Introduction

Febrile infection-related epilepsy syndrome (FIRES) is a unique catastrophic, refractory epilepsy syndrome that has attracted attention in recent years. FIRES affects previously healthy children aged 3–15 years and has unknown pathogenesis. It is defined by refractory status epilepticus (SE) that begins between 2 weeks and 24 h after a febrile illness, with or without fever at the onset of SE [1,2,3]. The disease is rare, with an estimated incidence of 1 in 1,000,000 and a prevalence of 1 in 100,000 [3]. The prognosis for FIRES is poor, with 12% of cases dying in the acute phase and 93% of survivors developing drug-resistant epilepsy (DRE) and significant cognitive impairment [4,5,6]. A third of all epilepsy patients without a specific epilepsy syndrome, such as FIRES, develop DRE, which is complemented by a deterioration in the quality of life and cognitive dysfunction [7]. Additionally, DRE increases the risk of suicide and sudden unexpected death in epilepsy (SUDEP), accounting for 7–17% of deaths in patients with epilepsy and up to 50% in those with refractory epilepsy [8,9]. Recent genetic studies have identified candidate genes for SUDEP that support the emerging concept of cardiocerebral channelopathy [10,11].

Although the pathogenesis of FIRES remains unclear, non-specific febrile infections occur within 2 weeks prior to the onset of refractory SE, which might cause an inflammatory cascade that may be involved in the pathogenesis of FIRES [4]. A significant elevation of inflammatory cytokines in patients with FIRES suggests an immune-mediated process [12,13,14,15]; thus, immunomodulatory therapies, including high-dose steroids, intravenous immunoglobulin (IVIG), and plasmapheresis have become the mainstream treatment [16]. However, evidence on the effectiveness of these therapies is still poor [16]; alternative interventions, such as tacrolimus, cyclophosphamide, rituximab, hypothermia, ketogenic diets, and neuromodulation have also been attempted [2]. Despite these special treatments, results regarding intractable epileptic syndromes, including FIRES, have not been satisfactory to date.

Evidence of a connection between neuroinflammation and epilepsy has been established using both experimental animal models and human studies [17,18,19]. Various cytokines have been recognized as potentially related to the pathogenesis of epilepsy [18,20]. In particular, a member of the interleukin (IL)-1 gene family, IL-1β, has been suggested to be associated with epileptic pathogenesis. IL-1β is a master cytokine that drives both brain and systemic inflammation, which activates a ubiquitous cell surface receptor, IL-1 receptor type 1 (IL-1R1). The IL-1β–IL-1R1 axis activates a cascade of inflammatory molecules, including other cytokines and chemokines. An IL-1 receptor antagonist (IL-1RA) is an endogenous competitive antagonist of IL-1R1 that blocks the activity of IL-1β (Figure 1).

There is emerging evidence that IL-1RA exerts potent anticonvulsant effects via selective inhibitors [21,22,23], and the possibility that it can inhibit epileptogenesis has also been postulated [24,25]. The IL-1 signaling pathway has been implicated in the pathogenesis of human epilepsy based on surgically resected epileptogenic foci from patients with DRE [19,26,27] [28,29]. Recent studies, including our own, have demonstrated the correlation between IL-1β and IL-1R1 levels and the severity of epilepsy [30,31,32].

Anakinra (Kineret, Swedish Orphan Biovitrum [Sobi], Stockholm, Sweden) is a human recombinant form of IL-1RA, the first IL-1-targeting agent introduced in 1993. Since then, anakinra has dominated therapies targeting IL-1 in autoinflammatory and autoimmune diseases with pathological cytokine involvement [33]. In 2016, Kenney-Jung et al. reported the first case in whom anakinra was successfully used to treat a 32-month-old girl with FIRES [13]. In the same year, Jyonouchi et al. described the efficacy of anakinra, not only in suppressing seizures, but also in improving cognitive function, in four cases of refractory epilepsy other than FIRES [34]. Since then, anakinra has been used in some refractory epilepsies [35,36]. Evidence of the effect of anakinra in individuals with FIRES is increasing [13,37,38,39,40,41]. Moreover, a recent proposal by the FIRES working group also recommended the early use of anakinra [42].

However, anakinra is not effective in all cases, and much is unknown regarding its suitability and how it should be administered. Therefore, in this review, we sought to summarize the scientific and clinical literature related to the use of anakinra in patients with FIRES and DRE.

## 2. Materials and Methods

This narrative literature review was based on clinical case reports on the use of anakinra. A literature search was conducted using the PubMed database and included articles published up to April 2021. The keywords used in the search were “epilepsy” and “febrile infection-related epilepsy syndrome (FIRES)” AND “interleukin 1 receptor antagonist protein” OR “interleukin 1 receptor antagonist protein” OR “anakinra.”

## 3. Clinical Findings

Several case reports of FIRES (five single cases, a series of two cases, and a series of 25 cases) [13,37,38,39,40,41,43] and some refractory epilepsies (two single cases and a series of four cases) [34,35,36] treated with anakinra have been published. Their main characteristics are summarized in Table 1, Appendix A, and Table 2.

### 3.1. Anakinra for FIRES

#### 3.1.1. Case Study

In 2016, the first case of the use of anakinra for successful treatment of a 32-month-old girl with FIRES, unresponsive to anesthetic agents, midazolam, and phenobarbital, was reported by Kenney-Jung et al. [13]. With prolonged treatment with anakinra and multidisciplinary therapy, the patient showed no developmental or cognitive impairment, except for rare focal seizures.

Additionally, seven cases, ranging from 32 months to 21 years of age, were presented in case reports. Anakinra was administered at 2.5–20 mg/kg/day for 6–42 days after the onset of FIRES [13,37,38,39,40,41]. Anakinra was started from 5 days to 18 months after the onset of seizures and was mostly administered after various treatments, including anesthetics, various anticonvulsants, and anti-immunotherapy. A marked anticonvulsant effect was observed in five cases, and no effect was observed in the remaining two cases [38,40].

#### 3.1.2. Cohort Study

In 2020, an international cohort study of 25 patients with FIRES who were treated with anakinra was published [43]. The patients’ median age was 8 years (5.2–11 years), and all were treated with anesthetic agents to achieve seizure control prior to the initiation of anakinra. Anakinra was started at a median of 20 days (14–25 days) after the onset of seizures, with an initial median dose of 3.8 mg/kg per day (3–5 mg/kg per day) and a final median dose of 5 mg/kg per day (4–9 mg/kg per day). A seizure reduction of >50% was observed in 11 out of 15 patients (73%). The authors stated that there was no statistically significant difference due to the small sample size, but the overall neurological prognosis was poor. A trend toward a longer period of mechanical ventilation, intensive care unit (ICU) and hospital length of stay was noted in those without as compared to those with a seizure response.

In the evaluation of the Pediatric Cerebral Performance Category (PCPC), six patients had no or mild impairment (PCPC1-2), six had moderate impairment (PCPC3), and five had severe impairment or were in a vegetative state (PCPC4‒5). All surviving children developed DRE. Three patients (12%) died because of withdrawal of support due to persistent super-refractory SE and expected poor neurological outcomes. In this study, early anakinra initiation was associated with a shorter period of mechanical ventilation, ICU and hospital length of stay, and possibly with seizure reduction [43].

#### 3.1.3. FIRES Cases Refractory to Anakinra

From the case series, two of seven patients were refractory to anakinra; one patient responded effectively to tocilizumab, an IL-6 receptor antagonist. The patients exhibited elevated IL-6 levels in the cerebrospinal fluid (CSF) and were switched to tocilizumab. The patient had behavioral dysregulation and inattention, but the seizures apparently improved [40]. There were also other reports of cases in whom tocilizumab was effective for FIRES [45,46,47]. The other patient was resistant to various treatments, including deep brain stimulation of the centromedian thalamic nuclei (CMN-DBS), and had fallen into a vegetative state [38].

In a cohort study of 15 patients with available seizure frequency data, four patients did not demonstrate a 50% or greater reduction in seizures after 1 week of anakinra treatment. It may be difficult to conclude that these four cases were resistant to anakinra, given the short evaluation period. In rheumatoid diseases, the response to anakinra is usually assessed within 4–12 weeks of starting treatment [48]. However, in the comparison between the groups with and without 50% or greater seizure reduction, the anakinra-refractory groups showed a longer duration of mechanical ventilation and ICU length of stay, but this difference was not significant. The prognosis at discharge was moderate disability, severe disability, vegetative state, and death.

In a combined case series and cohort study, six patients (aged 4–9 years) were included in the uncontrolled epilepsy group with anakinra and were reviewed for reference. Interestingly, all six cases were male, and the prognosis was not favorable. Anakinra was started from 6–42 days after seizure onset, and the dose ranged from 3.8 to 20 mg/kg per day. There was no consistency in seizure types in the anakinra-refractory cases.

Since the suspected anakinra-refractory cases varied in terms of the timing and dose of anakinra administration, it was not possible to determine a consistent trend in refractory cases in the small number of cases studied. Whether sex differences in boys played a role in the response to anakinra cannot be determined, since FIRES shows a slight male preponderance [1,2,4,49]. However, based on the reported data, anakinra-refractory cases may have had a poor prognosis.

#### 3.1.4. Adverse Events of Anakinra

In clinical reports of seven cases, case 1 developed a drug reaction with eosinophilia and systemic symptoms (DRESS), which did not occur when anakinra was re-administered [13]. No other significant adverse effects were reported. In a cohort study of 25 cases, three children (12%) experienced DRESS, all of whom recovered after receiving corticosteroids, without complications. Two children (8%) developed cytopenia, which eventually resolved without special intervention. A total of 10 children (40%) developed infections after treatment with anakinra, but treatment was discontinued in only one case due to the infection.

### 3.2. Anakinra for DRE

A total of six DRE patients treated with anakinra and aged 6–13 years (male:female = 4:2) were presented. Anakinra was administered 1–8 years after the epileptic seizure at a dose of 1.5 3 mg/kg/day or 100 mg/day [34,35,36]. All patients had resolution or improvement in seizures, and five of the six patients also showed improvement in cognitive function [34,35,36].

Although behavioral symptoms (disturbed sleep and hyperactivity) improved with anakinra in patients with encephalopathy with electrical status epilepticus in sleep (ESES), the ESES pattern persisted. The ESES findings improved by additional treatment with sirolimus (an mTOR inhibitor) [36], which has been shown to be effective for treatment-resistant seizures in patients with tuberous sclerosis [38,50].

Baseline gene expression analysis in patients with generalized epilepsy revealed significant activation of gene pathways suggestive of systemic immune activation, such as Rap1 signaling [35], an upstream regulator of IL-1β production by the NLRP3 inflammasome [51]. These signs nearly vanished with the resolution of epileptic seizures after anakinra treatment [35].

Despite the variety of the clinical diagnoses of epilepsy, a significant anticonvulsant effect has been observed in cases who do not show improvement with antiepileptic drugs, vagus nerve stimulation, or immunologic therapy. As for seizure types, anakinra was effective against unknown onset tonic‒clonic and generalized onset absence seizures. The cost and uncertainty of how long anakinra should be continued may be problematic. No apparent adverse reactions were observed.

## 4. Scientific Findings

### 4.1. Cytokine Analysis of Clinical Cases

The first case treated with anakinra showed that the CSF levels of the proinflammatory cytokines, IL-8 (from 4523 pg/mL before treatment to 15 pg/mL after treatment) and IL-6 (from 252 pg/mL to 4 pg/mL), decreased after the injection of anakinra, although this could be due to steroid pulse treatment and other factors. High levels of IL-8 and IL-6 were not observed in the serum [13]. Similarly, in a 6-year-old boy with FIRES, high CSF levels of IL-6 (84 pg/mL, reference < 25 pg/mL) and IL-8 (605 pg/mL, reference < 205 pg/mL) were detected during the treatment, but the serum levels were within normal limits [40]. Although some patients showed elevated pro-inflammatory cytokine and neopterin, an inflammatory biomarker, levels in the CSF or serum in the cohort study, there was no indication of the time of measurement or measurement values [40].

Measurement of IL-1β is also critical, as anakinra is an antagonist of IL-1β; however, measuring IL-1β levels is not easy [14,52,53]. A systematic review of various studies of cytokines other than IL-1β yielded controversial results [20], suggesting difficulties in the assessment of cytokine levels in human studies. The timing of seizures, specimen processing, and blood draws can affect cytokine levels [54,55,56,57,58]. In particular, circulating IL-1β is highly unstable, and thus levels of IL-1β often appear to be within the normal range, despite highly active systemic autoinflammatory diseases in most conventional assays. This may be due to its existence in macrovesicles or due to its short half-life [14,52,59].

Analysis of IL-1β levels was attempted using a standard cytometric bead array or enzyme-linked immunosorbent assay, but this was deemed unreliable [13,53]. In the cases included in this review who were treated with anakinra, there were limited reports of elevated IL-1β levels, although the method of measurement was not described. Recently, Clarkson et al. detected the elevation of IL-1β in a cell-based reporter assay [14]. Evaluation of circulating IL-1RA from patients with FIRES showed less effective blocking of IL-1β signaling than IL-1RA from healthy controls [14]. These results indicate that IL-1 RA inhibitory activity is functionally insufficient in patients with FIRES [14]. Although circulating serum IL-1β was not measured, flow cytometry analysis (FACS) assessment of cytokine production or levels of intracellular cytokines of monocytes and other cells in patients with intractable epilepsy has been suggested to be associated with clinical symptoms [32,34,36,60]. Our recent FACS analysis of patients with DRE demonstrated that they had higher levels of intracellular (peripheral monocytes), but not serum IL-1β compared to controls, which correlated with the frequency of seizures [32]. Thus, in vitro studies using FACS can detect IL-1β and allow subsequent analyses.

### 4.2. IL-1RA-Mediated Regulation of Epileptogenesis

The fact that even patients who survive FIRES develop intractable epilepsy is an important issue. Rodent models have shown the possibility of suppressing epileptogenesis through the IL-1β–IL-1RA axis [24,61], although mounting evidence suggests that IL-1RA exerts potent anticonvulsant effects via selective inhibitors [21,22,23]. A current experimental febrile animal model has shown that a rapid and transient increase in IL-1β, caused by prolonged febrile seizures not but simple seizures in early life increased seizure susceptibility in adults via IL-1R1. A single injection of exogenous recombinant IL-1RA could dose-dependently and time-dependently reverse this enhanced adult seizure susceptibility [61]. Notably, IL-1RA was effective only when administered within 24 h after the seizures, because changes in IL-1β occurred within 12 h after febrile induced seizures [61]. The FIRES proposal recommends early administration of anakinra [42] to protect against epileptogenesis; however, extremely early administration of anakinra might be warranted. It is also noteworthy that anakinra not only suppresses seizures but also improves cognitive function in patients with long-standing, intractable epilepsy [34,35]. Earlier administration of anakinra may be more effective, but delayed administration might also be worthwhile, as also described in the FIRES proposal [42].

In candidate gene analyses of FIRES, there was a significant association between the number of tandem repeats of the RN2 allele of IL-1RN and FIRES. The variation in this allele was associated with higher levels of IL-1β and lower levels of IL-1RA. A potential imbalance of intrinsic functional deficiency in endogenous IL-1RA and active IL-1β could lead to an unopposed pathological inflammatory state [62]. In an analysis of patients with febrile seizure, patients with prolonged seizures showed a trend for higher IL-1β and lower IL-1RA levels in blood samples than in controls [63]. However, febrile seizure patients without SE showed a trend for normal to lower levels of IL-1β and a trend for higher levels of IL-1RA [64,65]. Nevertheless, sampling may be a problem: specimen collection is not standardized and varies from subject to subject [63]. However, clinically, the risk of simple febrile convulsions transitioning to epilepsy is quite limited, which is consistent with the fact that epileptogenesis associated with convulsion superposition can be constructed. Seizure duration is an important determinant of epileptogenesis [66]. Therefore, it is essential to administer exogenous IL-1RA immediately to block IL-1 signaling in both clinical and experimental situations. IL-1RA might prevent future epileptogenesis due to febrile seizure superimposition by blocking this process [61]. Similarly, suppression of intractable and long-lasting seizures by anakinra might be effective in the treatment of intractable epilepsy that may develop as sequelae.

### 4.3. Potential Indicator of Anakinra Administration

While a single early dose of anakinra might be sufficient to suppress epileptogenesis in patients with prolonged SE, it may be necessary to administer sufficient doses early and for a longer period in patients with refractory, long-lasting seizures, such as FIRES. In fact, there have been cases of patients with FIRES whose seizures worsened again after stopping anakinra, based on EEG indicators [34].

It is unclear how long anakinra should be continued, and the indicators for anakinra use are unclear. In some anakinra refractory cases, anakinra treatment was relatively short (5 or 9 days) [43]. For comparison, response to anakinra in rheumatic diseases has been assessed within 4–12 weeks of therapy [48]. Naturally, the pathology of these conditions is markedly different, and cost may have been an issue.

Using FACS, Jyonouchi et al. demonstrated the IL-1β level and IL-1β/IL-10 ratio for peripheral blood monocytes in response to anakinra for clinical seizures and administered anakinra based on monocyte cytokine production in patients with intractable epilepsy [34,36]. FACS not only detects IL-1β but may also provide a guide for determining the duration of anakinra treatment. However, cytokines are also affected by infection and other factors, and thus the effects of other comorbidities need to be considered when interpreting the results [34].

There is a need for prolonged IL-1RA treatment, and a switch from anakinra to canakinumab, which has a longer half-life, should be considered to reduce the burden on the patient, as daily subcutaneous injections can be burdensome for patients. Canakinumab has the longest half-life (23–26 days) of the three available anti-IL agents, reducing the number of injections required [67]. DeSena et al. showed that once-a-month canakinumab can improve patients’ quality of life [35].

## 5. Conclusions

In this review, we present the findings that anakinra is effective for treating FIRES and refractory epilepsy. Thirty-two patients with FIRES were treated with anakinra, and of the 22 patients with available data, 16 patients (73%) had at least short-term seizure control. Anakinra led to marked improvements in some cases [13,37,39]. Functional deficiency of IL-1RA and decreased levels of intracellular IL-1RA were observed, providing direct evidence of the mechanism underlying the therapeutic effect [14]. However, FIRES remains an ill-defined severe epileptic syndrome that probably has multiple etiologies and pathogenic mechanisms, and multiple pro-inflammatory cytokines are potentially involved in the pathogenesis of FIRES [12,15,68]. We describe six suspected anakinra-refractory cases with poor prognoses, in whom the cause of the anakinra-refractoriness is unclear. Switching from anakinra to an IL-6 receptor antagonist suppressed seizures in a patient, allowing school attendance despite behavioral dysregulation and inattention [40]. The role of cytokine confirmation tests has not been established, and the treatment of FIRES should not be delayed for the analysis of cytokines [42].

Delays in diagnosis and appropriate treatment have been identified as a cause of poor prognosis in patients with FIRES [2], and a cohort study showed that early administration of anakinra may improve prognosis [43]. Early diagnosis and prompt treatment of FIRES are undoubtedly important, but a definitive diagnosis in the acute phase is not always straightforward. Therefore, empiric therapy must be considered. The recently proposed FIRES recommendations also suggested that presumptive treatment of FIRES should ideally be initiated within 1 week of the initial diagnosis, once other generally treatable causes have been ruled out with reasonable certainty [42]. In the absence of an established treatment for FIRES, anakinra, which has no serious side effects, is one of the options. It is optimal to start anakinra before or within 2 weeks after the first seizure [42].

Patients who experience seizures early in life are more likely to develop epileptic pathology later in life, as expressed in the phrase “seizures beget seizures” [69]. Early anakinra administration may be a promising approach, even for intractable epilepsy. However, although there has been a small number of cases, anakinra treatment of intractable epilepsy has been shown to be quite effective, even after long-term treatment, with marked recovery and improvement in cognitive function [34]. It may thus be suitable to consider anakinra before invasive treatment.

However, daily subcutaneous injections are quite burdensome for these patients, and if their efficacy is confirmed, consideration should be given to switching to canakinumab, which has a longer half-life. Nevertheless, because no serious adverse reactions have been reported to date, anakinra may become a new tool for epilepsy treatment.

In the future, a large-scale prospective study is needed to understand the optimal timing, dosage, and duration of anakinra treatment. Furthermore, the rational biological correlates of the anakinra response, as well as its safety and efficacy should be determined; and biomarkers, such as cytokines, should also be measured and correlated with clinical symptoms as much as possible.

Although it is not currently considered an established treatment, anakinra has the potential to become a new tool in the treatment of epilepsy.

## Figures and Tables

**Figure 1 ijms-22-06282-f001:**
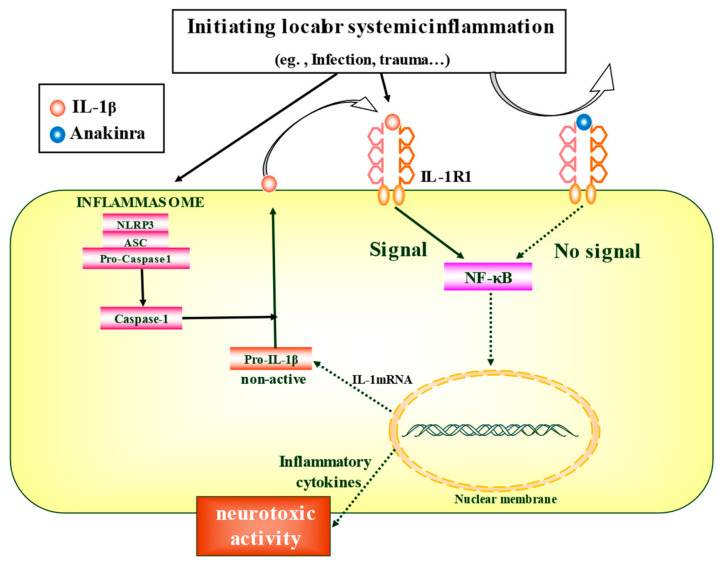
Mechanism of action of anakinra. Initiating local or systemic inflammation triggers, such as infections, can mediate the formation of the inflammasome, consisting of ASC, NLRP3 and pro-caspase-1. Pro-caspase-1 released by the formation of inflammasomes self-digests each other to become activated Caspase-1, which cleaves and matures the precursor of IL-1β. The activation of IL-1β though type I IL-1 receptor, which ultimately activates the transcription factor NF-κB, which stimulates the production of inflammatory cytokines and leads to the inflammatory cascade. Anakinra blocks IL-1 receptor 1, antagonizing the effects of IL-1β and exerts neuroprotective or anticonvulsant effects by blocking these neuroinflammatory cascades. ASC: apoptosis associated speck-like protein containing caspase activation and recruitment domain, IL: interleukin, IL-1-R1: interleukin-1 receptor 1, NF-κB: nuclear factor kappa-light-chain-enhancer of activated B cells, NLRP3: NOD-Like Receptor containing pyrin domain 3.

**Table 1 ijms-22-06282-t001:** Clinical features and Treatment of Febrile Infection-Related Epilepsy Syndrome refractory to anakinra.

Case * /Ref.	Onset Age (Years)/Sex	ASM	Special Additions	Sz Onset to Anakinra (Days)	Anakinra Dose (mg/kg/Day)	Anakinra Duration (Days)
4/[44]	5/M	PB, MDZ, TH, KETA, CBD	KD CMN-DBS	22	titrated up to 10 mg/kg/day	90
6/[40]	6/M	LEV, PHT, PB, MDZ, KE, PENT, VPA, TPM, CBD, LZP	IVIg, steroid, Tocilizumab, KD	6	titrated to 20 mg/kg/day	15
9/[43]	9/M	MDZ, PENT, LID, CBD, DBS	IVIg, steroid, PE, KD	42	4 mg/kg/day	>114
10/[43]	5/M	MDZ, PENT, LID, KE, CBD, DBS	IVIg, steroid, PE, KD	21	10 mg/kg/day	>124
24/[43]	8/M	MDZ, PENT, CBD	IVIg, steroid, HYPO	6	3.8 mg/kg/day	9
32/[43]	4/M	MDZ, PENT, KE, CBD	IVIg, steroid, PE, rituximab	33	5 mg/kg/day	5

Abbreviations: M, male; ASM, anti-seizure medication; CBD, cannabinoids; HYPO, hypothermia; KE, Ketamine; IVIG, intravenous immunoglobulin; ISO, isoflurane; KD, ketogenic diet; LFT, liver function test; MDZ, midazolam; PENT, Pentobarbital; PE TH, Thiopentone, plasmapheresis; Sz, seizure; PRP, Propofol; OXC, Oxcarbazepine. Case * The numbers of the cases presented in Appendix A. Special additions* Immune therapy including IVIg, steroid, or CMN-DBS, KD, HYPO, PE.

**Table 2 ijms-22-06282-t002:** Clinical features and treatment of drug-resistant epilepsy.

Case/Ref.	Clinical Diagnosis of Epilepsy	Onset Age (Years)/Sex	ASM */Special Additions	Sz Onset to Anakinra (Years)	Anakinra Dose (mg/kg/Day)	Clinical Findings	Developmental Prognosis
1/[34]	(1) focal onset impaired awareness (2) unknown onset tonic -clonic	13/M	IVIG, VNS	4	2 mg/kg/day → 4 mg/kg/day	decreased grand mal seizures from 1–2x/week to 1x/3–4 weeks	More alert, expressive, and attentive, with improved social interactions and three-dimensional vision.
2/[34]	(1) focal onset impaired awareness (2) unknown onset tonic -clonic	8/F	IVIG, Steroid	2	3 mg/kg/day	seizure-free for 1 year while receiving 100 mg of anakinra. When reduced to 75 mg, cluster seizures occurred.	ND
3/[34]	Landau-Kleffner syndrome	6/M	IVIG, Steroid, VNS	4	3 mg/kg/day	Thalidomide and VNS reduced seizure frequency until discontinuation due to adverse reactions. Seizures resolved with addition of anakinra for over 4 years	Improvement in motor skills and cognitive skills. Photophobia and persistently dilated pupils also resolved.
4/[34]	ND	9/M	IVIG, Steroid	1	1.5 mg/kg/day	decreased grand mal seizure once a year	Improved cognitive skills (more attentive and focused), reduced irritability, and hyperactivity.
5/[35]	Generalized onset absence	14/F	dexamethasone	ND	100 mg daily → 100 mg/day twice daily	100 mg of anakinra once daily, 80% reduction in seizure frequency 100 mg twice daily, no clinically evident seizures for two months.	Profound improvements in her fatigue, general malaise, quality of life, and academic performance
6/[36]	ESES (nocturnal seizure)	6/M	IVIG, Steroid, mTOR	2	100 mg/day	reduced behavioural symptoms, but not the ESES pattern (N60% SWI in all areas). Adding Sirolimus for 7 weeks improved the ESES pattern on EEG.	Hyperlexia and echolalia remained

Abbreviations: ASM *, anti-seizure medication. All patients were treated with multiple antiepileptic drugs; ESES, encephalopathy with electrical status epilepticus in sleep; mTOR, mammalian target of rapamycin; VNS, vagal nerve stimulation; IVIG, intravenous immunoglobulin.

## Data Availability

The datasets generated and/or analyzed during the current study are available at the PubMed database repository (https://pubmed.ncbi.nlm.nih.gov/).

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
