# Peer review of "Towards a Treatment for Neuroinflammation in Epilepsy: Interleukin-1 Receptor Antagonist, Anakinra, as a Potential Treatment in Intractable Epilepsy"

_ijms, 2021, doi:10.3390/ijms22126282_

Round 1

Reviewer 1 Report

This is a quite interesting and detailed narrative review on the effectiveness of anakinra in febrile infection-related epilepsy syndrome (FIRES) and other drug-resistant epilepsies. This review is very helpful, giving a new and encouraging perspective of treatment for intractable epileptic syndromes.

Literature data are adequately reported and updated. Tables and Supplementary file provide complete information on the clinical and EEG findings of patients.

I have not particular queries.

Author Response

Journal IJMS (ISSN 1422-0067)

Manuscript ID ijms-1250422

Type Review

Number of Pages 15

Title: Towards a treatment for neuroinflammation in epilepsy: Interleukin-1 receptor antagonist, anakinra, as a potential treatment in intractable epilepsy

Reviewer comment: 1
This is a quite interesting and detailed narrative review on the effectiveness of anakinra in febrile infection-related epilepsy syndrome (FIRES) and other drug-resistant epilepsies. This review is very helpful, giving a new and encouraging perspective of treatment for intractable epileptic syndromes.

As readers might regard these treatments as an established treatment, I could not recommend this article for publication.

Literature data are adequately reported and updated. Tables and Supplementary file provide complete information on the clinical and EEG findings of patients.

I have not particular queries.

Response

I would like to thank you for taking the time to review my paper. We appreciate your time and effort once again.

Reviewer 2 Report

Comments

Summary:

This is a narrative review article about anakinra and DRE. The authors systematically reviewed the literatures by some key words related to FIRES and anakinra.

Major issues:

The authors systematically reviewed case reports. This methodology is not narrative, but systematic review. However, the number of literatures were small and contents ware close to narrative way.

In my understanding, narrative review is requested from editors or someone who is well experienced writes to disseminate important information of well-established treatment. Since the Anakin is not a gold standard for FIRES or as anti-seizure medications, the contents of this narrative review article is too soon to say so far.

As readers might regard these treatments as an established treatment, I could not recommend this article for publication.

Author Response

Journal IJMS (ISSN 1422-0067)

Manuscript ID ijms-1250422

Type Review

Number of Pages 15

Title: Towards a treatment for neuroinflammation in epilepsy: Interleukin-1 receptor antagonist, anakinra, as a potential treatment in intractable epilepsy

To the Reviewer 2

We appreciate the time and effort that you have dedicated to provide your insightful feedback. We have revised our manuscript according to your comments, as much as possible. All suggested revisions, as well as additional ones to improve the language of the manuscript, are indicated in underline and yellow text. We sincerely hope that, with these revisions, our manuscript will be suitable for publication in your esteemed journal.

Reviewer comment 1

The authors systematically reviewed case reports. This methodology is not narrative, but systematic review. However, the number of literatures were small and contents ware close to narrative way.

Response 1

At first, we would like to share the definition of “reviews” with reviewers. This is because the word "review" can mean different things to different people. The definition we are adopting is based on the Grant et al definition1.

According to this definition, systematic review is a format that is often conducted based on the Cochrane Collaboration or the NHS Centre for Reviews and Dissemination. As you pointed out, our search strategy was reproducible and systematic. However, in the subsequent quality assessment of the articles, we did not use a method that eliminates bias and it does not meet the Cochrane criteria. Therefore, while the search method is systematic, our paper as a whole is not classified as a systematic review. The fact that only case reports were included is also different from systematic reviews, which usually use randomized control trials to obtain conclusive evidence.

 Therefore, we took the form of a literature (narrative) review. We therefore took the form of a literature (narrative) review because, as the reviewer said, anakinra is not yet a gold standard. We are confident that this review will give us a better understanding of the situation and help us to identify directions for future research. We also believe that a literature review is useful in this exploratory context.

  • Grant MJ, Booth A. A typology of reviews: an analysis of 14 review types and associated methodologies. Health Info Libr J. 2009 Jun;26(2):91-108. doi: 10.1111/j.1471-1842.2009.00848.x. PMID: 19490148.

Reviewer comment 2

In my understanding, narrative review is requested from editors or someone who is well experienced writes to disseminate important information of well-established treatment. Since the Anakin is not a gold standard for FIRES or as anti-seizure medications, the contents of this narrative review article is too soon to say so far.

Response 2

We agree with your comment that ‘A narrative review is requested from editors or someone who is well experienced writes to disseminate important information of well-established treatment’. On the other hand, there is also a form of narrative review called ‘narrative Overview’ which does not follow this format. Please refer to the definition2) for details. In fact, this Journal ‘International Journal of Molecular Sciences’, seems to have accepted narrative review that it did not invite. Thus, we took the form of a ‘narrative review’.

Once again, anakinra is not the golden standard though anakinra is mentioned in the FIRES proposal. However, we hope that as soon as possible, those involved in epilepsy care will become aware of the benefits of anakinra. For this reason, we believe that this narrative review provides a valuable insight into the status of anakinra as a new treatment candidate for FIRES and DRE.

Please note that the abstract and conclusions have been amended to avoid any misunderstanding that anakinra is an established treatment.

  • Grant MJ, Booth A. A typology of reviews: an analysis of 14 review types and associated methodologies. Health Info Libr J. 2009 Jun;26(2):91-108.

The following corrections have been made to the abstract (Page 1, Line 29-30)

Before

‘Anakinra may likely become a new tool for epilepsy treatment.’

to

‘Anakinra has the potential to become a new tool for intractable epilepsy treatment, but currently it does not have a solid evidence base.’

The following has been inserted at the end of the Conclusion (Page 10, Line 367-378)

‘Anakinra has the potential to become a new tool in the treatment of intractable epilepsy, but it is not currently a solid treatment’

Reviewer 3 Report

This is a well written narrative review concerning neuroinflammation in epilepsy.

I would recommend the authors to improve the background on the topic and to review also the implication of events related to sudden death during febrile seizures. I would suggest to include some references regarding the role of genetics inducing SCD in epilepsy:   

1. Genetic investigation of sudden unexpected death in epilepsy cohort by panel target resequencing by Coll M, et al  Int J Legal Med. 2016 Mar;130(2):331-9. doi: 10.1007/s00414-015-1269-0. Epub 2015 Sep 30. 

1. Genetic and forensic implications in epilepsy and cardiac arrhythmias: a case series Partemi S, et al Int J Legal Med. 2015 May;129(3):495-504. doi: 10.1007/s00414-014-1063-4. Epub 2014 Aug 15. PMID: 25119684

Author Response

Journal IJMS (ISSN 1422-0067)

Manuscript ID ijms-1250422

Type Review

Number of Pages 15

Title: Towards a treatment for neuroinflammation in epilepsy: Interleukin-1 receptor antagonist, anakinra, as a potential treatment in intractable epilepsy

To the Reviewer 3

We appreciate the time and effort that you have dedicated to provide your insightful feedback. We have revised our manuscript according to your comments, as much as possible. All suggested revisions, as well as additional ones to improve the language of the manuscript, are indicated in underline and yellow text. We sincerely hope that, with these revisions, our manuscript will be suitable for publication in your esteemed journal.

Reviewer: 3
Comments to the authors

This is a well written narrative review concerning neuroinflammation in epilepsy.

I would recommend the authors to improve the background on the topic and to review also the implication of events related to sudden death during febrile seizures. I would suggest to include some references regarding the role of genetics inducing SCD in epilepsy:  

  1. Genetic investigation of sudden unexpected death in epilepsy cohort by panel target resequencing by Coll M, et al Int J Legal Med. 2016 Mar;130(2):331-9. doi: 10.1007/s00414-015-1269-0. Epub 2015 Sep 30.
  2. Genetic and forensic implications in epilepsy and cardiac arrhythmias: a case series Partemi S, et al Int J Legal Med. 2015 May;129(3):495-504. doi: 10.1007/s00414-014-1063-4. Epub 2014 Aug 15. PMID: 25119684

Response

Thank you very much for your meaningful remarks.

We made the following insertion according to your comment (Page2, Line 47-48).

Recent genetic studies have identified candidate genes for SUDEP that support the emerging concept of cardiocerebral channelopathy [10, 11].

Round 2

Reviewer 2 Report

Okay, I understand the explanation that was provided by the authors. 

Author Response

Journal IJMS (ISSN 1422-0067)

Manuscript ID ijms-1250422

Type Review

Number of Pages 15

Title: Towards a treatment for neuroinflammation in epilepsy: Interleukin-1 receptor antagonist, anakinra, as a potential treatment in intractable epilepsy

Reviewer comment: 1

Okay, I understand the explanation that was provided by the authors.

Response

We thank you for your understanding of our explanation.

 Once again we would like to acknowledge the referees who took the time to review our paper and provided thoughtful comments.